# Environmental Assessment with Cage Exposure in the Neva Estuary, Baltic Sea: Metal Bioaccumulation and Physiologic Activity of Bivalve Molluscs

Nadezhda Berezina [1], Alexey Maximov [1,*], Andrey Sharov [2,3], Yulia Gubelit [1] and Sergei Kholodkevich [2]

1   Zoological Institute, Russian Academy of Sciences, Universitetskaya Embankment, 1,
    199034 St. Petersburg, Russia; nadezhda.berezina@zin.ru (N.B.); gubelit@list.ru (Y.G.)
2   St. Petersburg Federal Research Centre, Russian Academy of Science, 14th Line Vasilievsky Ostrov 39,
    199178 St. Petersburg, Russia; sharov_an@mail.ru (A.S.); kholodkevich@mail.ru (S.K.)
3   Papanin Institute for Biology of Inland Waters of the Russian Academy of Sciences, 152742 Borok, Russia
*   Correspondence: alexeymaximov@mail.ru

**Abstract:** The rise in anthropogenic impacts on the marine environment requires new water management. The use of a triadic approach (bioaccumulation, bioassay, and physiological biomarkers) has been shown to have good applicability for the comparative assessment of the environmental state of the Neva Estuary (Gulf of Finland, Baltic Sea). The novelty of the methodological approach of the study was that it involved both active and passive bio-monitoring methods for assessing the quality of estuarine environment. The classical analyses of metal concentration in bottom sediments, in field biota (fish and molluscs), and in caged molluscs were accompanied by a bioassay of sediment toxicity using amphipods. The physiological state of molluscs kept in cages was assessed according to two functional characteristics, such as cardio-tolerance and metabolic activity (oxygen consumption rate), after exposition in cages. The method of active monitoring (caging exposure with molluscs) as a measurement of parameters in clean molluscs has proven itself well in controlling the accumulation of both metals and oil products. Macroalgae that are abundant in estuarine ecosystems are also good indicators of metals, at least showing the transition from bottom sediments to the next level of food webs. Unionid molluscs were found to be a more sensitive and effective indicator of contaminant accumulation than dreissenid molluscs, characterized by a low tolerance to changeable environmental conditions in the estuarine ecosystem and rather high mortality in cages. Our results have shown that caging exposure with unionids can be a widely used methodological approach for the assessment of estuarine environmental quality through the determination of metal concentrations in molluscs and their physiological state.

**Keywords:** heavy metals; zebra mussel; pond mussel; oxygen consumption; cardioactivity; bioassay; bioaccumulation factor; Gulf of Finland

## 1. Introduction

The rise in anthropogenic impacts on the marine environment requires the development of new water management and environmental protection approaches. Estuaries are ecologically important habitats and are considered transitional ecosystems between the ocean and rivers [1]. Estuarine systems are subject to anthropogenic pressure (from industry, cities, tourism, and agriculture), vulnerable to these impacts, and are among the most threatened aquatic environments [2,3]. To develop effective methods for the assessment and monitoring of pollution in estuaries, it is necessary to identify and recommend bioindicators and indices sensitive to the prevalent type of pollution that has regionally specific features.

According to modern ideas and world experience, assessing the ecological and toxicological state of the aquatic environment is no longer possible without bioindication

and data on the integral toxicity of substrate using biotesting (bioassay). Bioassay is a procedure for establishing the toxicity of the environment using test objects that signal danger, regardless of which substances and in what combination cause changes in their organism's functions. Due to its rapidity and accessibility, biotesting is widely used around the world, along with methods of analytical chemistry.

The selection of sensitive test objects and their state indices, reflecting the relationship between the quality of the environment and the health of a living organism, is also relevant in the case of the Baltic Sea coastal areas. In this region, chemical pollution (heavy metals, oil pollution) and hypoxia of bottom waters are among the main factors responsible for the deterioration of habitat quality [4,5]. Besides natural sources (erosion of rocks, volcanic emissions), heavy metals (arsenic, cadmium, copper, mercury, lead, zinc) are released into the environment in large quantities from activities associated with mining, metallurgy, manufacture, fossil fuel combustion, or waste incineration. Oil product pollution is also prevalent in this estuary due to active shipping and port development, so the selected indicators and their indices should be sensitive to both types of contaminants.

The levels of metals were mostly determined in fish and molluscs, which showed high positive correlations with the levels of metals in the environment [6]. The consumption of molluscs and fish contaminated with toxic metals and other toxicants as marine products can show adverse effects on humans [7]. At present, relatively little is known about the biological effects of hazardous substances of different types on the physiological indices of benthic animals, including molluscs [5]. At the same time, being directly associated with water and bottom sediments, benthic animals can indeed become one of the best indicators. Heavy metals accumulate in the tissues of aquatic animals through ingestion or by passing through semi-permeable membranes [8], such as animal gills and skin.

Metals such as copper (Cu) and zinc (Zn) are essential for animal metabolism, while others such as mercury (Hg), cadmium (Cd), and lead (Pb) have no known role in the regulation of biological processes [9]. At the same time, they can pose carcinogenic and neurodegenerative disorders in animals organisms [10]. Arsenic (As), which is also not an essential element for organism functioning but is found in some marine environments and accumulated in the tissues of fish, is a highly toxic and carcinogenic environmental pollutant [11]. The toxicity of heavy metals (As, Hg, Cd, and Pb) depends largely on their chemical forms (inorganic or organic), exhibiting varying toxicity levels [12].

The toxicity of metals in the environment relates to their concentration, form, availability, and accumulation by organisms. In general, the inorganic form is more toxic than the organic form. Although quite a lot of modern articles are devoted to measuring the accumulation of these toxic metals in aquatic biota, they rarely consider how the accumulation of harmful substances is associated with physiological disorders in the animal organism [13].

The bioaccumulation factor (BAF) is an important indicator in environmental risk assessment because it provides quantitative information about the ability of a pollutant to be taken up by organisms from the environment and food. Therefore, the BAF factor is not constant, since its value depends on the concentration in the environment [14]. It is often used as one of the first screening parameters for persistent and toxic substances [15,16]. Aquatic macrophytes and macroalgae are widely used as biological quality elements for environmental assessment of coastal waters. Many papers exist that focus on macroalgae as indicators of pollution [16–18]. The level of accumulated pollutants in algal thalli (in mass units per g or kg of dry algal biomass) and bioconcentration or bioaccumulation factor serve as a basis for environmental monitoring and water quality assessment [17].

Using transplanted molluscs from a reference site in a polluted area can be a feasible strategy for biomonitoring the environment in coastal or estuarine zones. This is a rather novel approach that is widely tested [19,20]. Earlier, it was shown that contaminant levels in caged mussels increased rapidly and, in one month, achieved stable levels for various metals [21]. There are studies [22,23] that suggest zebra mussel, *Dreissena polymorpha* (Pallas, 1771), and representatives of Unionidae [24] as a substitute in brackish and fresh waters for bivalves *Mytilus*, which is widely used in marine environments. The first one

has a relatively smaller size, which can be a problem in terms of multiple analyses, while unionid molluscs are characterised by a relatively large size, which is essential in terms of performing several different analyses, including physiological tests and bioaccumulation. Penetrating into the tissues of aquatic animals, metals accumulate in them in higher concentrations than in the environment, leading to the replacement of essential elements with toxic ones and a negative effect on animals [25].

We have carried out a comparative assessment of the state of the environment in the Neva Estuary (Gulf of Finland) using physiological biomarkers, biotesting, and bioaccumulation. The novelty of the methical approach of the study was the application of both active and passive biomonitoring methods for assessing the quality of the estuarine environment. The classical analyses of metal concentration in bottom sediments, in field biota (fish and molluscs), and in caged molluscs were accompanied by a bioassay of sediment toxicity using amphipods. The physiological state of molluscs kept in cages was assessed by two functional characteristics, such as cardio-tolerance and metabolic activity (oxygen consumption rate), after exposition in cages. It was necessary to find out whether the new methodological approach (i.e., the use of a triadic method: biotesting with crustaceans, bioaccumulation indices, and the physiological state of transplanted molluscs) will effectively determine the correlation between pollution and the consequences of its impact on biota. Bioassay and bioelectronic methods were used prior to testing the impact of various hazardous substances on the state of aquatic organisms [26–28], while the triadic approach and assessment of bioaccumulation in caged molluscs will be applied for the first time. The results obtained can become the basis for choosing the most effective indicators in biota and their further use of the cage method in the monitoring of heavy metals and environmental quality.

Field-testing of the physiological state of bivalve molluscs based on a bioelectronic cardiac system for the purposes of environmental monitoring was applied in the Gulf of Finland (Baltic Sea) in the 2010s [20,26,27]. In this system, the test animals applied are included directly in the system structure as primary converters; thus, they are an integral part of an electronic recording system of certain physiological and behavioural parameters reflecting the integrated response of animals to changes in environmental conditions [29]. To identify changes in the functional state of the mollusc organism caused by the action of unfavourable environmental factors, the animals were subjected to stress in the form of standardised short-term tests [26]. The recovery time of heart rate in molluscs and heart rate variability (deviation from norm) in this case are used as physiological biomarkers of the adaptability of the cardiac system to stress (cardio-tolerance).

## 2. Materials and Methods

### 2.1. Study Area

The estuary of the Neva River is located in the easternmost part of the Baltic Sea (Gulf of Finland). It is relatively shallow (average depth: 38 m), and salt content ranges from 0.10 to 6 g/L, decreasing from west to east. The Neva River flows out of Lake Ladoga and annually discharges 76 km$^3$ of fresh water into the headwaters of the gulf. In the gulf, there are three large coastal bays, Vyborgsky Bay, Luga Bay, and Koporskaya Bay, and several ports, including two large oil terminals (near Primorsk and Sovetsk) and newly constructed coal ports (Ust-Luga and Bronka).

The coastal zone of the eastern part of the Neva estuary (Gulf of Finland) is intensively used for various industries, including recreation (Resort District of St. Petersburg) and other uses (Leningrad Nuclear Power Plant, Waste Water Treatment). Large-scale dredging was previously carried out in Luga Bay and is currently being carried out in Neva Bay in the area of the construction of the Bronka Port and the creation of new artificial territories in St. Petersburg. Dredging results in increased turbidity and the re-release of buried chemicals from bottom sediments. In recent decades, most of the wastewater from St. Petersburg has been treated before being discharged into the bay. However, secondary water pollution from historical discharges persists (in the 1970s and 1990s). In summer

2019–2020, the content of chlorophyll-a (Chl-a) and the total phosphorus (TP) in water of the euphotic zone of the estuary varied from 1 to 19.2 μg/L and from 41 to 75 μg/L, respectively [4]. Chl-a concentrations in the estuary are closely related to changes in TP concentrations and the amount of precipitation in the summer, which is directly translated into changes in surface water runoff and Neva River flow [30]. Thus, eutrophication is a typical threat to the ecosystem of the Gulf of Finland. Among chemicals, heavy metals are the most widely distributed pollutants in the coastal area, while deepwater is characterised by a higher accumulation of organotins (TBT) and polyaromatic hydrocarbons [5,31].

### 2.2. Sampling Design

Observations in the field were conducted in the period July–August of 2020 and 2021 in relation to the state research topic of ZIN RAS and HAZLESS Project on twelve sites in the coastal zone (depths around 1–6 m) of the Neva estuary, eastern Gulf of Finland (Figure 1 and Table 1). Active biomonitoring methods such as cage exposure with molluscs were tested at six sites (St 1, 2, 6, 9, 10, and 11). Passive biomonitoring was also conducted along the Neva estuary coast. Field molluscs and fish were collected at six sites (St 3, 4, 5, 7, 8, and 12) where cages were not established. At all the twelve study sites, we collected water samples (0.25 mL) for the detection of total phosphorus, oil products, water salinity (salt content), and the upper 3 cm level of sediment for bioassay using the amphipod crustacean *Gmelinoides fasciatus* survival rate and chemical analyses (metals, PAHs, organic carbon). Macrozoobenthos (organisms with body size >2–3 mm) in the Neva estuary as a zone of critical salinity is initially poor; as a result, not all stations can provide sufficient biomass of biota for multi-analysis of pollutants and their effects, so the use of cages is especially important in such conditions.

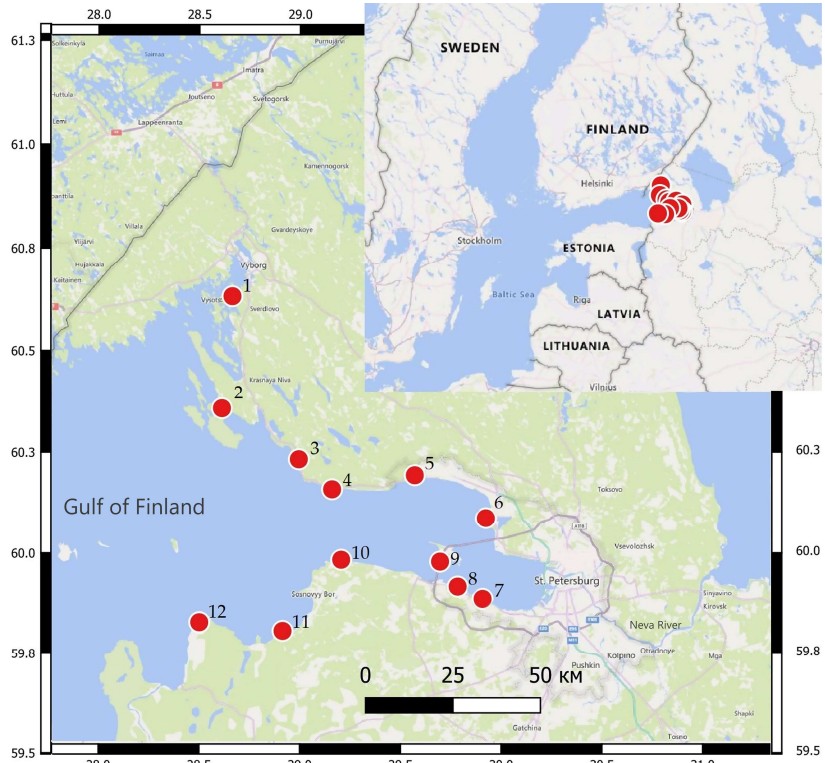

**Figure 1.** Sampling sites (St. 1–12) in the Neva River estuary, the eastern Gulf of Finland.

**Table 1.** Study sites characteristics. H is depth (m), Sal is water salinity (g/L), TP is total phosphorus (µg/L), THC is total concentration of hydrocarbons (mg/L), C org is % concentration of organic carbon in sediment, 16 PAHs is sum of the concentration of 16 polyaromatic hydrocarbons in sediment (µg/kg), Ma is mean dry mass of macroalgae g/m$^2$ (coefficient of variation %), and Surv is amphipod survival rate (%). GES and BES mean good and bad environmental state according to bioassay.

| No. | Name | Method | Latitude Longitude | H | Sal | TP | THC | Corg | 16PAHs | Ma | Surv | State |
|---|---|---|---|---|---|---|---|---|---|---|---|---|
| 1 | Zimino | Caged molluscs Sediment, Fish | 60.6348 28.6620 | 4 | 1.7 | 25 | 0.016 | 0.3 | 64 | <0.1 | 90 | GES |
| 2 | Primorsk | Caged molluscs Sediment, Macroalgae | 60.3605 28.6097 | 6 | 2.7 | 34 | 0.02 | 0.5 | 595 | 87 (10) | 60 | Sub-GES |
| 3 | Okunevaya | Sediment, Fish | 60.2352 28.9913 | 4 | 2.5 | 25 | 0.008 | 0.45 | 49 | <0.1 | 90 | GES |
| 4 | Flotsky | Sediment, Fish, Macroalgae | 60.1613 29.1563 | 1 | 1.2 | 25 | 0.012 | 0.5 | 91 | 97 (31) | 90 | GES |
| 5 | Serovo | Sediment, Field molluscs, Macroalgae | 60.196 29.567 | 1 | 0.8 | 27 | 0.01 | 0.8 | 57 | 35 (52) | 80 | GES |
| 6 | Dubki | Caged molluscs, Sediment, Fish | 60.0896 29.9195 | 2 | 0.3 | 50 | 0.011 | 0.8 | 49 | 47 (33) | 95 | GES |
| 7 | Petergof WPT | Sediment, Field molluscs, Macroalgae | 59.8888 29.9033 | 1 | 0.2 | 43 | 0.009 | 0.9 | 250 | 19 (55) | 65 | Sub-GES |
| 8 | Lomonosov beach | Sediment, Field molluscs, Macroalgae | 59.9193 29.7796 | 1 | 0.4 | 47 | 0.01 | 0.8 | 65 | 15 (40) | 70 | Sub-GES |
| 9 | Dam, Port | Caged molluscs, Sediment | 59.9812 29.6922 | 5.5 | 0.67 | 25 | 0.024 | 0.96 | 1033 | 10 (30) | 60 | Sub-GES |
| 10 | Grafskaya | Caged molluscs, Sediment, Macroalgae | 59.9858 29.2014 | 3 | 2.6 | 48 | 0.023 | 0.45 | 725 | 80 (17) | 20 | BES |
| 11 | River Sista mouth | Caged molluscs, Sediment, Macroalgae | 59.8092 28.9104 | 2 | 3.2 | 30 | 0.032 | 0.59 | 314 | 17 (30) | 80 | GES |
| 12 | Luga Bay | Sediment, Macroalgae | 59.8309 28.4969 | 1 | 3.8 | 25 | 0.008 | 0.29 | <16 | 67 (32) | 90 | GES |

Adult individuals of the bivalve molluscs, *Unio pictorum*, and *D. polymorpha* were taken from the reference site, located in a pristine place of the Rybinsk Reservoir (salt content 0.4 g/L) without any contamination (58.0653° N, 38.2555° E; see Figure 1). Overall, 90 individuals of *Unio* (55 to 60 mm) and 300 individuals of *Dreissena* (20–22 mm) were selected for caging. All the selected animals looked healthy and active. Experimental cages were closed cube-shaped cages made of a strong frame with internal partitions. They were covered with a net with a mesh diameter of 2 mm (Figure S1). Dreissenid molluscs were placed inside the cage on the top shelf, and unionid molluscs were placed inside the cage on the bottom shelf. Cages containing native molluscs (15 individuals of unionid molluscs and 50 individuals of dreissenid molluscs; see Figure S2) from the reference site were placed near the bottom at six sites of the Neva estuary: three northern sites (from 28 July to 27 August 2020) and three southern sites (from 26 July to 27 August 2021). The exposure period for caged molluscs was 1 month (30 days), after which analyses of the metal accumulation and testing of the mollusc physiological indices were performed. The duration of the experiment was chosen based on previous results [25] to obtain a significant level of metal bioaccumulation. The month of experiment was selected as a temperature- and food-comfort period for both species of molluscs since bioaccumulation of pollutants depends on temperature and trophic conditions [21].

In addition, in the area of all study sites, annual macroalgae (*Ulva intestinalis* and *Cladophora glomerata*), which are widespread in the Baltic Sea [16,32]), were collected to estimate metal bioaccumulation. The mid-summer macroalgal bloom observed every year in the Neva estuary and in other eutrophied marine ecosystems is due to the development of high biomasses of opportunistic filamentous macroalgae. After storms and windy weather, macroalgae mats, detached from hard substrates, accumulate at the bottom of the coastal zone, often causing hypoxic conditions and other unfavourable eutrophication phenomena [33–35]. The biomass of macroalgae was evaluated at all sites where they were present during the survey in July 2021, and samples of algal tissue were taken for bioaccumulation of metals and oil products. To evaluate biomass, macroalgae were sampled using a cylindrical metal frame (a 0.8 m height and a sampling area of 0.03 m$^2$) in three replications per site. The algae were collected from the frame, detaching hard substrates (if needed), and washed with fresh water to remove associated fauna. In the laboratory, algae samples were dried in air to a constant weight and weighed to 0.01 g precision. Their biomass was estimated as a mean dry weight (DW) per 1 m$^2$.

### 2.3. Biotesting

The 10-day survival tests with amphipods were used for the quality assessment of sediments. This species was shown to be a useful test species in various regions, including coastal areas of the Baltic Sea [28]. The sediments (3 cm of the upper layer) were collected at each study site via bottom grab (Van Veen). Amphipods *G. fasciatus* were collected from the natural populations in the Neva estuary, near St 6 (Dubki Park, Sestroretsk) and transported in isothermal containers to the laboratory. They were used in a 10-day survival test; the procedure was previously described in detail [28]. Briefly, the test sediment was sieved through a 0.25 mm sieve prior to testing to eliminate the ingress of other invertebrates. After a 3-day period of media equilibration, 20 individual test animals were introduced. During the validation experiments, the water in the test beakers (0.5 L) was gently aerated with an aquarium pump through the tip of a pipette fixed 2 cm above the sediment. As a result of this experimental procedure, the concentration of dissolved oxygen and pH were maintained at the same level from the beginning to the end of exposure. Amphipods were fed daily with a 1:3 mixture of fish food and dried seaweed (TetraMin®, Company Tetra Spectrum Brands Pet, LLC 3001 Commerce St., Blacksburg, VA, USA). After a 10-day exposure period, the overlying water was carefully decanted, and the sediment was sieved through a 0.5 mm sieve to retain the amphipods, which were then washed in a glass dish, counted, and visually analysed for activity. The survival of amphipods in each variant was calculated as the percentage of living individuals at the end of the exposure in relation to the initial number. The ecological state of the studied areas can be classified on a three-point scale: good environmental state (GES, survival of amphipods is in the range of 70–100%), sub-GES (amphipod survival ranges from 69 to 50%), and bad environmental state (BES, amphipod survival < 49%) [28].

### 2.4. Chemical Analyses

Chemical analyses of bottom sediments and biota were carried out in an accredited analytical laboratory (OOO "LABORATORIYA", http://ecolabspb.ru/ accessed on 11 August 2023). Before analysis, all samples of bottom sediments were dried in an oven at 30 °C and sieved through a plastic sieve with a pore diameter of 1 mm. The passing fraction was crushed in an agate mortar and split using a combination of ultrapure acids HCl/HF/HNO3 (1:1:1) in a Mars 5 microwave oven (CEM, USA, https://cem.com accessed on 11 August 2023). Cleavage products were transferred to polypropylene vials and then diluted to 50 mL with deionized water.

The concentrations of metals/metalloids cadmium (Cd), copper (Cu), zinc (Zn), and arsenic (As) in bottom sediments and soft tissues of animals and macroalgae was analysed via inductively coupled plasma optical emission spectrometry (ICP-OES) on an iCAP6300 series spectrometer according to the M-MVI 80-2008 method. Mercury was determined via

the flameless AAS method on a RA-915+ mercury analyser (manufactured by «LUMEX», https://www.lumex.com/ accessed on 11 August 2023). Measurement accuracy was controlled using a certified standard CRM 5365-90 (OOO "LABORATORIYA", http://ecolabspb.ru/ accessed on 11 August 2023) and provided a suitable recovery (<5%).

Analysis of PAH compounds in sediments and biota mainly involves extraction with organic solvents, purification, and separation by high-performance liquid chromatography (HPLC) with ultraviolet light. Analysis of the content of polycyclic aromatic hydrocarbons (PAH) was carried out via HPLC using the method FR.1.31.2004.01279. Altogether 16 PAHs were detected including Benzo-a-pyrene, Anthracene, Acenaphthene, Acenaphthylene, Benzo-a-anthracene, Benz-β-fluorantene, Benz-k-fluorantene, Benz(-g,h,i-)perylene, Dibenz[a,h] anthracene, Indeno (1,2,3-cd)pyrene, Pyrene, Naphthalene, Phenanthrene, Fluoranthene, Fluorene, and Chrysene. The content of organic carbon in bottom sediments (C org, % of the dry mass of sediments) was determined by the loss on ignition method, the limit was 0.03–9.99%. The total phosphorus in water was measured by use of the spectrophotometer PE-5300B. The oxygen content in water was determined with a HANNA oxygen-meter and varied between 7.0–7.2 mg/L at all study sites.

### 2.5. Bioaccumulation Factor

The bioaccumulation factor (BAF) of metals by bivalve molluscs was calculated as the ratio of the average metal concentration in the body (Co, mg/kg d.w.) and its average concentration in bottom sediments (Cs, mg/kg d.w.) according to the formula:

$$BAF = Co/Cs.$$

The concentrations of metals in bottom sediments were recalculated to the 1% level of organic carbon, and the concentrations in animal tissues were recalculated per dry weight, assuming that the dry weight of animals is 20% of wet weight [36].

### 2.6. Mollusc Physiological State

After cage exposure, two species of bivalve molluscs, U. pictorum and *D. polymorpha*, were carefully transported (in isomeric bags, T = 17–18 °C) to the laboratory to test their health state. The second species has high mortality at two sites (St 6 and 10; see Table S2); therefore, we tested it at fewer stations. At least eight individuals of each species and each site were taken for one test of cardioactivity and twelve individuals to measure the metabolic rate of animals.

The metabolic (energy) rate was evaluated by measuring the oxygen consumption in caged dreissenid molluscs. The oxygen consumption rate (OCR) of zebra mussels was measured in 100 mL closed respirometers with a wide neck. Clean flasks were filled with filtered water through a tube, and one individual was placed there, with the lid tightly closed, preventing air bubbles from entering the water when closing. OCR was calculated as the difference between the oxygen levels in the experiment and control (without animals) after 4 h of exposure; before calculating the average, all indicators led to the same mass of the molluscs. Dissolved oxygen was measured using a Hanna HI 9142 oxygen metre (https://hannacan.com/ accessed on 11 August 2023) calibrated before the start of measurements according to the standard.

During testing of cardioactivity, we evaluated two indices, such as heart rate (HR) and HR recovery time after physical loading. In the case of unionid molluscs, the load was a short-term increase in water salinity, while in the case of dreissenid molluscs, it was a short-term transfer from water to air. The original fibre-optic method was applied [37] to measure heart rate. The pulsation of heart muscle in mollusc modulated periodical changes in the light reflection and dispersion of a low-intensity laser semiconductor. The optical signal coming from the sensor attached to the surface of the mollusc shell was converted to a digital signal using the photoplethysmograph and the original software VarPulse 10.0 (RIC "EcoContour" Co., Ltd., St. Petersburg, Russia, https://ecocontour.ru accessed on

11 August 2023) (Figures S3 and S4). Mean HR values (bites/min) were registered after the rate stabilised at the constant level for at least 1 h.

The recovery of heart rate (HRR, min) of unionid molluscs after 1 h of water salinity increases up to a concentration of 6 g/L by adding NaCl solution [27]. HRR of *Dreissena* was carried out on the basis of the analysis of their reactions to the short-term (60 min) exposition on air. After 1 h of loading, the molluscs were returned to their previous conditions. HRR was estimated as the time in minutes needed to restore HR to its level before loading. The quality criteria of HRR were established for unionid molluscs (HRR < 70 min testify GES and >100 min testify BES, see [27]) and until they are not clear for *Dreissena*. Therefore, we compared the obtained values of HR and HRR in both species with their values in molluscs from the reference site.

*2.7. Statistics*

Variables are presented as mean values, standard errors, and 95% confidence intervals. We conducted the pairwise Mann–Whitney tests after a significant Kruskal–Wallis test (analogue of ANOVA for non-parametric H-statistics). Relationships between variables had been analysed with Spearman rank correlation and via principal component and classification analysis. Multivariate comparisons were used to group parameters and sampling sites according to their characteristics. The data were log-transformed, centred, and normalised to avoid misclassification due to differences in data dimensions. Zero mean and unit variance (z-scores) were normalised by subtracting their mean from each value and then dividing by their standard deviation. These z-scores were further used in correlation matrices in multivariate analysis (including PCA). Software: Past 4.13 and Statistica 10.0 were used to analyse data.

**3. Results**

*3.1. Physical and Chemical Parameters of Sediment Statistics*

Salinity, depth, and sediment toxicity are presented in Table 1, and concentrations of metals and PAHs are in Table 2. Salinity increased from east to west, with a decrease in the influence of the Neva River from 0.2 to 3.2 g/L. The total phosphorus concentration (TP) ranged from 25 to 48 µg/L, with the highest value at St 10 (Grafskaya Bay) on the southern coast of the Neva estuary. Four study sites (St 2, St 7, St 8, St 10) ranged within values that characterised "eutrophic conditions" with nutrient-rich water (TP > 30 µg/L, see Table 1).

**Table 2.** The concentration of metals in bottom sediments (mg/kg d.w.) and in biota (µg/g d.w.). MAC is a maximum admissible concentration according to regional normative criteria. Coefficient variation was less than 20% for all measurements (*n* = 3). Concentrations of metals over the MAC range are written in bold.

| Metal Metalloid | St 1 | St 2 | St 3 | St 4 | St 5 | St 6 | St 7 | St 8 | St 9 | St 10 | St 11 | St 12 | MAC |
|---|---|---|---|---|---|---|---|---|---|---|---|---|---|
| | | | | | | Sediment | | | | | | | |
| Cd | **1.6** | 0.38 | 0.12 | 0.18 | 0.2 | 0.2 | 0.39 | 0.42 | 0.39 | 0.16 | <0.05 | 0.24 | 1 |
| Pb | **42** | **13** | 2.9 | 7.9 | **11** | 5 | 2.2 | 7.8 | 8.7 | 4.2 | 1.5 | 1.7 | 10 |
| Zn | **250** | 33 | 16 | 24 | 26 | 26 | 20 | 33 | 34 | 10 | 7.7 | 4.2 | 199 |
| Cu | 46 | 7.1 | 1.6 | 4.4 | 6.7 | 5.1 | 20 | 12 | 10 | 1.3 | 1.4 | 0.84 | 50 |
| Hg | 0.005 | 0.032 | 0.17 | 0.022 | 0.005 | 0.005 | 0.09 | 0.08 | 0.032 | 0.005 | 0.005 | 0.005 | 0.05 |
| As | 0.05 | 0.05 | 0.05 | 0.05 | 0.05 | 0.05 | 0.05 | 0.20 | 0.05 | 0.05 | 0.05 | 0.38 | 0.5 |

**Table 2.** *Cont.*

| Metal Metalloid | St 1 | St 2 | St 3 | St 4 | St 5 | St 6 | St 7 | St 8 | St 9 | St 10 | St 11 | St 12 | MAC |
|---|---|---|---|---|---|---|---|---|---|---|---|---|---|
| | | | | | | Molluscs | | | | | | | |
| Cd | 4.7 | 2.7 | - | 0.4 | 2.6 | 2.2 | 2.1 | 2 | 0.8 | 1.8 | 1.9 | 2.1 | 5.00 |
| Pb | 0.92 | 0.68 | - | 1.2 | 6.9 | 0.68 | 2.5 | 1.7 | 1.1 | 0.1 | 3.1 | 1.7 | 7.50 |
| Zn | 150 | 108 | - | 14.8 | 47 | 116 | **200** | **170** | 25 | 5 | 23 | 21 | 170 |
| Cu | 5.3 | 5.2 | - | 3 | 10.2 | 3.4 | 20 | 6.9 | 1.6 | 0.56 | 8.8 | 2.8 | 50 |
| Hg | 0.001 | 0.001 | - | 0.05 | **0.1** | **0.14** | 0.06 | 0.04 | 0.05 | 0.04 | 0.06 | 0.001 | 0.1 |
| | | | | | | Macroalgae | | | | | | | |
| Cd | - | 3.6 | - | 0.4 | 2.6 | 4.9 | 0.18 | **5.6** | 3.5 | 1.3 | 1.4 | 1.6 | 5.00 |
| Pb | - | 10 | - | 1.02 | 6.9 | 3.2 | <0.1 | 1.0 | 22 | 5.9 | 2.1 | 1.1 | 7.50 |
| Zn | - | 45 | - | 12.8 | 47 | 27 | 7.9 | 16 | 40 | 28 | 23 | 11.1 | 170 |
| Cu | - | 11 | - | 3 | 10.2 | 5.6 | 0.37 | 3.9 | 8.7 | 7.9 | 5.8 | 3.8 | 50 |
| | | | | | | Fish | | | | | | | |
| Cd | **10** | **11** | **21** | 4 | **5.2** | - | 0.11 | 0.18 | - | - | - | - | 5.00 |
| Pb | 0.17 | 0.1 | 0.1 | 0.8 | 0.95 | - | 0.1 | 0.1 | - | - | - | - | 7.50 |
| Zn | 73 | 41 | 44 | 108 | 112 | - | 5 | 7.9 | - | - | - | - | 170 |
| Cu | 0.8 | 0.6 | 1 | 5 | 4.4 | - | 0.37 | 0.37 | - | - | - | - | 50 |
| Hg | **0.14** | **0.25** | **0.21** | **0.22** | **0.24** | **-** | **0.21** | 0.05 | - | - | - | - | 0.1 |

Sediment consisted of silty sand at all locations and large stones. Organic carbon in soft sediments ranged from 0.3 to 0.96% varying insignificantly between sites. The environmental state based on the amphipod survival test of sediment toxicity was good at most sites (Table 1). We revealed the only case (St 10) with bad environmental conditions (amphipod survival was 40%). Spearman's correlation revealed a significant relationship between amphipod survival and total PAH concentration in sediment ($Rs = -0.84$), which in turn correlated positively with THC ($Rs = 0.78$) in water (Table S2).

### 3.2. Metal and PAHs Concentrations in Biota and Sediments

The concentration of metals (mg/kg) in sediments and soft tissues (mg/g d.w.) of animals (molluscs and fish) and macroalgae is presented in Table 3. St 1, 2, and 5 were distinguished by the highest concentration of Cd and Pb and Zn in sediment, and their values exceeded the maximum admissible concentration (Table 3). Molluscs from the Reference site contain low concentrations of metals (Cd 0.1, Pb 0.4, Cu 8.5, Zn 20, and Hg 0.05 µg/g d.w.), and no hydrocarbons were found in their tissue.

**Table 3.** BAF of various metals in molluscs and macroalgae from study sites.

| Metals | Cd | Pb | Zn | Cu | Hg |
|---|---|---|---|---|---|
| | | | Macroalgae | | |
| Mean | 25.5 | 24.2 | 3.1 | 3.8 | - |
| Standard Error | 7.6 | 8.5 | 0.8 | 1.6 | - |
| Median | 18.5 | 14.0 | 1.8 | 1.4 | - |
| Standard Deviation | 26.4 | 29.4 | 2.9 | 5.4 | - |
| Minimum | 0.5 | 0.2 | 0.4 | 0.0 | - |
| Maximum | 100.0 | 100.0 | 9.1 | 15.6 | - |
| 95% Confidence Level | 16.8 | 18.7 | 1.8 | 3.4 | - |

**Table 3.** *Cont.*

| Metals | Cd | Pb | Zn | Cu | Hg |
|---|---|---|---|---|---|
| | Unionid molluscs | | | | |
| Mean | 24.2 | 0.9 | 5.5 | 2.7 | 9.2 |
| Standard Error | 8.5 | 0.4 | 1.4 | 1.1 | 3.5 |
| Median | 14.0 | 0.3 | 5.3 | 1.2 | 2.2 |
| Standard Deviation | 29.4 | 1.3 | 4.8 | 3.9 | 12.1 |
| Minimum | 0.2 | 0.1 | 0.8 | 0.2 | 0.1 |
| Maximum | 100.0 | 3.5 | 17.2 | 11.5 | 35.0 |
| 95% Confidence Level | 18.7 | 0.8 | 3.1 | 2.5 | 7.7 |

Cd concentrations in molluscs at 58% of the sites were above 2 μg/g d.w., although in general, they are not above maximum acceptable values (MAC). Macroalgae also contained significant amounts of cadmium (>1 μg/g d.w.) at 70% of the sites, and it reached 5.6 μg/g d.w. at St 8, which is above the MAC. Exceedances of MAC for concentrations of Hg and Zn in molluscs tissues were also noted at some sites (Table 2).

The level of bioaccumulation of metals was associated with environmental factors (Table S2). For example, Unionid Zn decreased with increasing water salinity. The accumulation of Pb by unionids decreased with increasing depth of the station ($Rs = -0.67$), and Zh accumulation by macroalgae, on the contrary, increased with increasing a station depth ($Rs = 0.69$). Sediment Cu ($Rs = -0.66$), mollusc Zn ($Rs = -0.63$) and Corg ($Rs = -0.76$) were negatively related with water salinity. Macroalgal biomass was negatively correlated with algal Cd concentration ($Rs = -0.59$). In addition, the contents of various metals in sediments and biota, as a rule, positively correlated with each other.

The bioaccumulation factor (BAF) was determined from the concentrations of metals in the bottom sediments and in the tissues of the test animals and macroalgae. BAF in macroalgae and molluscs varied between metals (Kruskal–Wallis test: macroalgae $H = 20.31$, $p = 0.0001$; Mann–Whitney pairwise comparisons, Bonferroni corrected, confirmed differences between the following: Cd vs. Pb, $p = 0.001$; Cd vs. Zn, $p = 0.007$; Cd vs. Cu, $p = 0.004$; and molluscs $H = 22.38$, $p = 0.0001$, Cd vs. Pb, $p = 0.002$; Cd vs. Cu, $p = 0.02$; Pb vs. Zn, $p = 0.01$).

The maximum value of BAF in macroalgae was 100 for Cd, and less for Pb, Zn, and Cu (3.12, 9.11, and 15.6, respectively). The BAF in molluscs for Cd maximally reached 100 and an average of 24.2 (Table 3 and Figure S5). The BAF of Hg of molluscs also significantly greater than for other metals, averaging 9.2. Fish (percoid and cyprinid groups) accumulated generally higher amounts of Cd and Hg (Table 2), and especially high concentrations of Cd were found in fish from St 1–3.

Principal component and classification analysis (PCA) revealed three principal components (PCs) accounting for 21.15%, 19.04, and 16.35%, respectively, of the total variation in the dataset (Figure 2). They represented altogether over 56% of total variation. All three PC loadings are presented in Table 4.

The first component (PC1) has positive loadings for the variables, representing unionid concentrations of Zn and Cu and sediment Zn and Cu and has negative loadings for salinity, oil products (PAHs and THC) and macroalgae biomass. The second component, PC2, has positive loadings for the concentration of Hg in molluscs and has negative loading for the depth of stations and sediment concentrations of Cu, Zn, and Pb. The PC3 has positive loadings for the salinity, Cd concentration in molluscs, and amphipod survival rate and has negative loadings for the total phosphorus in water, organic carbon and Cu in molluscs (Table 4).

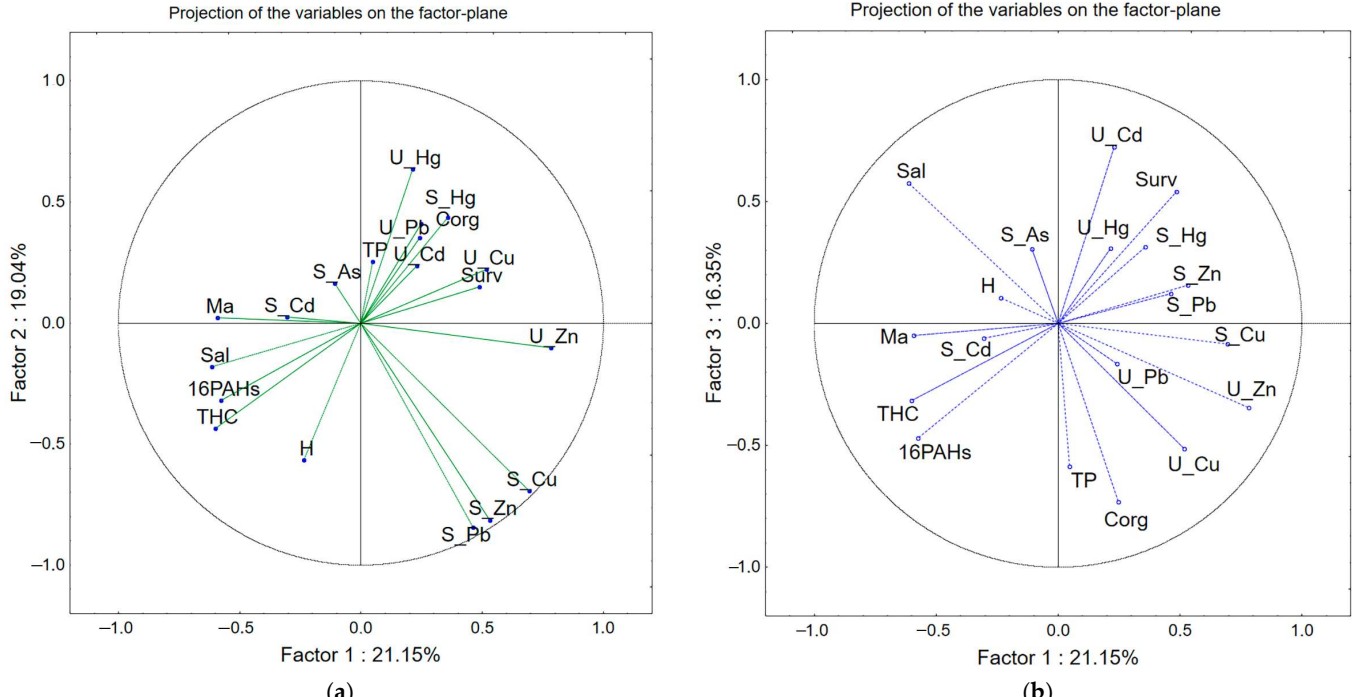

(**a**)　　　　　　　　　　　　　(**b**)

**Figure 2.** Projection of variables related with metal accumulation of unionid molluscs and site physical and chemical characteristics on the factor plane as a result of PCA: (**a**) Factors 1 and Factor 2; (**b**) Factors 1 and Factor 3. Diagram shows the median, the mean values, the lower and upper quartiles, and the minimum and maximum values of the BAF of different metals.

**Table 4.** Factor coordinates of the variables, based on correlations. Values in bold show significant correlation.

| Variable | Factor 1 | Factor 2 | Factor 3 |
|---|---|---|---|
| H | −0.24 | **−0.57** | 0.1 |
| Sal | **−0.61** | −0.18 | **0.58** |
| TP | 0.05 | 0.25 | **−0.59** |
| THC | −0.6 | −0.44 | −0.32 |
| Corg | 0.25 | 0.41 | **−0.73** |
| 16PAHs | **−0.58** | −0.32 | −0.47 |
| Ma | **−0.59** | 0.02 | −0.05 |
| Surv | 0.49 | 0.15 | **0.54** |
| S_Cd | −0.3 | 0.02 | −0.06 |
| S_Pb | 0.46 | **−0.84** | 0.12 |
| S_Zn | 0.53 | **−0.81** | 0.16 |
| S_Cu | **0.69** | **−0.69** | −0.09 |
| S_Hg | 0.36 | 0.44 | 0.31 |
| S_As | −0.11 | 0.16 | 0.31 |
| U_Cd | 0.23 | 0.24 | **0.72** |
| U_Pb | 0.24 | 0.35 | −0.17 |
| U_Zn | **0.78** | −0.1 | −0.35 |
| U_Cu | **0.52** | 0.22 | **−0.52** |
| U_Hg | 0.21 | **0.64** | 0.31 |

According to PCA and possible factors influencing the bioaccumulation of metals in molluscs, we could distinguish three sites (St 9, 10, and 2) as a separate group (highest concentration of hydrocarbons) that correlated negatively with two main principal components responsible for 40% of variance. These three sites can be characterized as a most eutrophied and contaminated locations by both metals and hydrocarbons. In addition, St 11, St 12, and St 4 form a closely related separate group; they correlated negatively with PC1 and are

similar to concentrations of Zn and Cu in sediment and biota and less contaminated than others. St 1 was significantly different of all other sites, negatively correlating with PC2 (i.e., highest concentration of Cd in molluscs). The rest sites (St 3, 5, 6, 7, and 8) also form one group, which related positively with all three principal components.

The highest amounts of PAHs in bottom sediments were at stations St 9 (1033 µg/kg), St 10 (725 µg/kg), and St 2 (595 µg/kg, see Table 1). The most abundant PAHs in the sediment were fluoranthene, fluorene, pyrene, benzo-a-pyrene, and benzo-a-anthracene (Table S3). The Spearman correlation analysis revealed significant relationships between concentration of PAHs in the bottom sediments and THC (0.78 µg/L) in water. PAH accumulation levels in molluscs were significant, especially in molluscs exhibited at St 8, 9, 10, and 11 (basically fluoranthenes, see Table 5). In molluscs at the reference site and at other stations, no oil products were detected (or they were detected in insignificant quantities).

**Table 5.** Concentrations of PAHs (mkg/kg) in biota (caged unionid molluscs and field molluscs and fish) at study sites.

| | Caged Unionid Molluscs | | | | | | |
|---|---|---|---|---|---|---|---|
| **PAH** | **St 1** | **St 2** | **St 6** | **St 9** | **St 10** | **St 11** | **Ref.** |
| Benzo-a-pyrene | <0.5 | <0.5 | <0.5 | <0.5 | <0.5 | <0.5 | <0.5 |
| Benzo-a-anthracene | <1 | <1 | <1 | <1 | <1 | <1 | <1 |
| Benz-$\beta$-fluorantene | <2 | <2 | <2 | 36 | 5.8 | 39 | <2 |
| Benz-k-fluorantene | <0.5 | 1 | 1.2 | 5.7 | 10 | 1.4 | <0.5 |
| Benz(-g, h, i-)perylene | <0.5 | <0.5 | <0.5 | 6.2 | <0.5 | 2.8 | <0.5 |
| Dibenz[a,h]anthracene | <1 | <1 | <1 | <1 | <1 | <1 | <1 |
| Indeno (1,2,3-cd) pyrene | <2 | <2 | <2 | <2 | <2 | <2 | <2 |
| Pyrene | <2 | <2 | <2 | <2 | <2 | <2 | <2 |
| Chrysene | <2 | <2 | <2 | <2 | <2 | <2 | <2 |
| | Field Fish and Molluscs | | | | | | |
| **Site** | **St 3** | **St 4** | **St 5** | **St 6** | **St 7** | **St 8** | **Ref.** |
| Benzo-a-pyrene | <0.5 | <0.5 | <0.5 | <0.5 | <0.5 | 0.62 | <0.5 |
| Benzo-a-anthracene | <1 | <1 | <1 | <1 | <1 | <1 | <1 |
| Benz-$\beta$-fluorantene | <2 | 1.5 | <2 | <2 | <2 | 38 | <2 |
| Benz-k-fluorantene | <0.5 | <0.5 | <0.5 | <0.5 | <0.5 | 10 | <0.5 |
| Benz(-g, h, i-)perylene | <0.5 | <0.5 | <0.5 | <0.5 | <0.5 | <0.5 | <0.5 |
| Dibenz[a,h]anthracene | <1 | <1 | <1 | <1 | <1 | <1 | <1 |
| Indeno (1,2,3-cd) pyrene | <2 | <2 | <2 | <2 | <2 | <2 | <2 |
| Pyrene | <2 | <2 | <2 | <2 | <2 | <2 | <2 |
| Chrysene | <2 | <2 | <2 | <2 | <2 | <2 | <2 |

### 3.3. Physiological State of Molluscs

We found up to 90% mortality after one month exposition of this molluscs at St 6 and 80% at St 10 (Table S1). At other sites, the survival of *Dreissena* was 60–80% (i.e., 30–40 ind.) with the lowest at St 8 (60%).

Metabolic rate (measured as an oxygen consumption rate) in *Dreissena* differed significantly between sites (H = 44.22, $p < 0.0001$, $n = 12$; Figure 3). Unlike *Dreissena*, in the case of Unionid molluscs, mortality was not observed. The highest oxygen consumption rates were recorded for unionid molluscs at St 9 (Dam) and St 10 (Grafskaya), and they were 66 and 38% higher than at the Reference location.

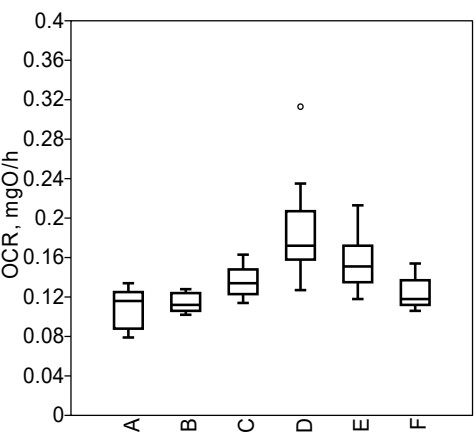

**Figure 3.** Mean SD and 95% confidence interval for oxygen consumption rate (OCR in *Dreissena polymorpha*. A is Reference, B is St 1, C is St 2, D is St 9, E is St 10, and F is St 11. Differences in OCR were significant (Kruskal–Wallis, H = 44.22, *p* < 0.0001, *n* = 12) between pairs AC (*p* = 0.048), AD (*p* = 0.01), AE (*p* = 0.03), BC (*p* = 0.01), BD (*p* = 0.0007), BE (*p* = 0.001), CD (*p* = 0.022), DF (*p* = 0.004), and EF (*p* = 0.033) according to Mann–Whitney test for pairwise comparisons, Bonferroni corrected.

Heart rate (HR) and their recovery time after loading (HRR) in molluscs *Dreissena* and *Unio* were different between sites (Figures 4 and 5). Significant decreases in HR of both species were found at St 2 compared to the Reference site and other sites. On the contrary, the second index (HRR) was much higher for this location in both species of molluscs than at the Reference site and other locations. In addition, unionid HRR values were 2.5–3.5 times significantly greater at St 9 and St 11 compared to the Reference site and St 1, 2, 6, and 10 (Figure 5). In accordance with the quality criteria for indicators of cardioactivity (HRR > 70 min), conditions at three stations 2, 9, and 11 were unsuitable for unionids (poor environmental status), which is associated with both increased salinity (St 2 and 11) and oil pollution (all three stations, Tables 1 and S3).

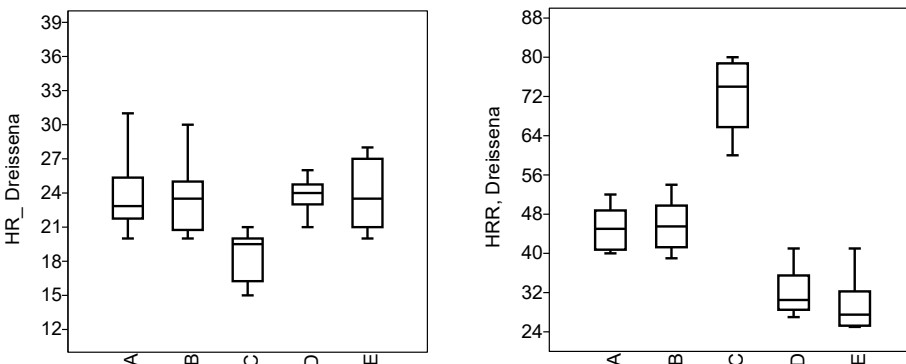

**Figure 4.** Mean SD and 95% confidence interval for HR and HRR of *Dreissena polymorpha*. A is Reference, B is St 1, C is St 2 (Primorsk), D is St 9 (Dam) and E is St 11 (Sista). Differences in HR were significant (Kruskal–Wallis, H = 15.72, *p* = 0.003) between pairs AC (*p* = 0.021), BC (*p* = 0.047), CD (*p* = 0.010), ED (*p* = 0.029). Differences in HRR were significant (Kruskal–Wallis, H = 32.48, *p* < 0.0001, *n* = 8) between pairs AC (*p* = 0.008), AD (*p* = 0.018), AE (*p* = 0.019), BC (*p* = 0.009), BD (*p* = 0.015), BE (*p* = 0.016), CD (*p* = 0.009) and CE (*p* = 0.009) according to Mann–Whitney test for pairwise comparisons, Bonferroni corrected.

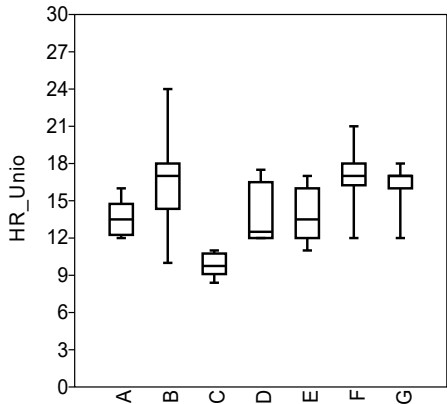 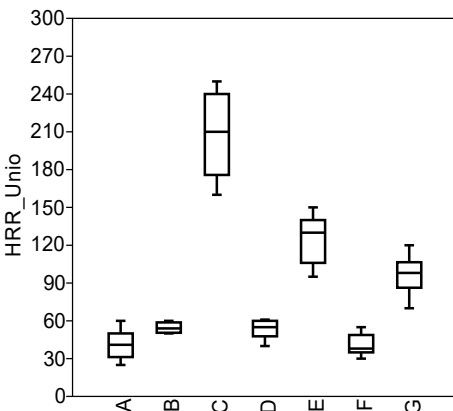

**Figure 5.** Mean SD and 95% confidence interval for HR and HRR of *Unio pictorum*. A is Reference site, B is St 1, C is St 2, D is St 6, E is St 9, F is St 10, and G is St 11. Differences in HR were significant (Kruskal–Wallis, H = 29.72, *p* < 0.0001, *n* = 8) between following pairs AC (*p* = 0.019), CD (*p* = 0.019), CE (*p* = 0.030), CF (*p* = 0.019) and CG (*p* = 0.019), DF (*p* = 0.041) and EF (*p* = 0.016). Differences in HRR were significant (Kruskal–Wallis, H = 47.73, *p* < 0.0001, *n* = 8) between pairs AB (*p* = 0.019), AC (*p* = 0.019), AE (*p* = 0.019), AG (*p* = 0.020), BC (*p* = 0.019), BE (*p* = 0.018), BG (*p* = 0.019), CD (*p* = 0.019), CE (*p* = 0.019), CF (*p* = 0.019), CG (*p* = 0.019), DE (*p* = 0.018), DG (*p* = 0.019), EF (*p* = 0.019), FG (*p* = 0.020), and CE (*p* = 0.009) according to Mann–Whitney pairwise comparisons, Bonferroni corrected.

## 4. Discussion

A comparative assessment of the state of the natural environment of the Neva Estuary (Gulf of Finland) using physiological biomarkers, biotesting and bioaccumulation showed good applicability of such a triadic approach. The method of active monitoring as a measurement of parameters in clean molluscs has proven itself well in both the control of the accumulation of metals and oil products. Although there is evidence that the accumulation of organic pollutants is slower than that of metals. In particular, it was shown that after half a year, the levels of PAHs and PCBs in the natural population were still higher than in caged mussels [21]. Under the conditions of the studied estuary, the levels of PAH accumulation through a monthly exposure were comparable in natural and caged unionids, and, on the whole, there were quite high. In the future, however, it is important to check the accumulation of organic substances in a longer experiment.

Macroalgae that are abundant in estuarine ecosystems are also good indicators of at least showing the transition from bottom sediments to food webs (they are the first link in this web). In a previous study, significant positive correlations between metal content in sediments, water, and algal tissues were obtained [16,38], which testifies to the high bioavailability of metals by macroalgae in conditions of the Neva estuary (Gulf of Finland). The high importance of macroalgae as indicators was shown, in [16,39] where authors used Environmental Quality Standard to assess pollution degree. High accumulative ability of opportunistic macroalgae together with their mass development makes macroalgae an important component for biogeochemical processes in the coastal zone [40]. Macroalgal mats concentrated at the bottom surface influence enzymatic activity of sediment microorganisms that directly affects biogeochemical cycles [40].

For the Baltic Sea, the background values of metals and their maximum admissible concentrations in the tissues of the studied animals have not yet been established, since there are still insufficient data. In this work, we considered regional standards (Table 2) that were obtained for fresh waters and that also require verification. Relating to standards from European regulations (EC No. 1881/2006), MAC values in biota for Pb were 1.5 µg/g ww, while for Cd it is 1.0 µg/g ww.

The distribution of metals in the coastal zone of the Gulf of Finland was well known as a result of previous studies [5,38,41]. In the area of river mouths, elevated levels of copper 10–12 µg/L and zinc 2–3 µg/L were determined [41]. Later studies [5] also identified

several areas in the easternmost part of the bay (estuary of the Neva River), in which the concentrations of cadmium (Cd), copper (Cu) and especially zinc (Zn) in water and bottom sediments above acceptable levels. For example, according to the maximum levels of copper and zinc in water were 0.10 and 0.46 µg/L, and in bottom sediments, 51 and 639 mg kg$^{-1}$ of dry matter. According to the Swedish Environmental Protection Agency [42] sediment pollution classification used in the Baltic Sea [41], the concentration range for Cd, Zn, and Cu at most stations exceeds the limits of quality classes 1 and 2, that is, no or slight pollution, being mostly within the 3rd class (significant pollution). The concentrations of metals in sediments (Cd = 0.3; Cu = 45; Zn = 95 mg/kg dry matter) were recognized as background for the Baltic Sea [41]. The average concentrations of metals in bottom sediments in the studied area of the Gulf of Finland for Cu were lower, and for Cd and Zn, 2 and 1.4 times higher than these background values. On them, the levels of metal contamination lie within the 3rd class of bottom sediment quality, showing moderate contamination. Within the class 3 quality of bottom sediments, the concentrations of Cd vary from 0.5 to 1.2 mg kg$^{-1}$, Cu—30–60 mg kg$^{-1}$, Zn—125–196 mg/kg d.w. [41].

The concentrations of metals in mollusc tissues obtained in the Gulf of Finland were similar to the range of those levels obtained by other authors for coastal ecosystems of different seas. In mussel *Mytilus galloprovincialis* from the Black Sea, the values of bioaccumulation of heavy metals varied within the following ranges: Cu 1.13–5.11 µg/g wet weight (or about five times more for dry weight); Cd 0.14–2.02 µg/g ww; Pb < 0.1–0.18 µg/g ww [43,44]. Concentrations of heavy metals in the other molluscs species (*Anadara*, *Mytilus*, *Rapana*) from the Romanian marine waters varied within wide ranges, as follows: Cu 0.93–12.9 µg/g ww; Cd 0.14–2.92 µg/g ww; Pb 0.01–0.37 µg/g ww. Also, for *Mytilus* from North Atlantic Ocean, the values were 0.88–1.93 (Cu), 0.07–0.57 µg/gww (Cd), 0.10–1.64 µg/gww (Pb) [45].

The values of metal accumulation are not constant, since they depend not only on the concentration of the metal in the environment, but also on the kinetics of its absorption and excretion from the body [14,46]. It is also known that Zn and Cu levels are regulated by organisms [46], in contrast to Cd, which can accumulate in organisms and be further transported and concentrated in food chains [47]. We found very high bioaccumulation factors for Cd and Hg in the conditions of studied estuary. Some authors also demonstrated high accumulation of metals, especially Cu, Cd, and Pb, by some species of Unionidae [48]. In this case, high Cu content in marine gastropods and bivalve species is normal since Cu is an essential element and is present in hemocyanin, the blood pigment of these invertebrates, which is vital for respiration and oxygen transport [49].

It is believed that the concentrations of Cd accumulated in organisms depend mainly on their level in the environment [50–52]. Under the studied conditions, direct correlations between specific metals and environmental factors were not found, which is more likely due to the conjugated influence of a number of factors and the relation with the eutrophication of the estuary. Environmental multifactor influence on Cd bioaccumulation were also found earlier [16,51,53]. When studying the relationship between metal content in bivalves and eutrophication, a statistically significant positive correlation of the metal pollution index with the level of eutrophication was confirmed [54], which means that eutrophication can affect the overall bioaccumulation of metals in molluscs. Given the complexity of this problem, it is critical to elucidate the underlying mechanisms of how coastal eutrophication affects metal bioaccumulation in bivalve molluscs.

Non-essential metals (lead, arsenic, mercury, and cadmium) are highly toxic, even at very low levels, especially if they accumulate in the metabolically active sites. The organism has to limit the accumulation of toxic metals or transform them into non-toxic forms. Toxic metals interfere with the normal metabolic functions of essential elements. Their binding to protein macromolecules causes disruption of normal biological function. Metal-catalysed formation of free oxygen radicals is involved in the production of many pathological changes, including mutagenesis, carcinogenesis, and ageing [50]. Therefore, molluscs with high concentrations of Cd and Hg can be a potential source of toxic metals for animals feeding upon them and contribute to further transfer and contamination in the

food chain. Concentrations of these two toxic metals (Cd and Hg) were particularly high in fish, partly due to the biomagnification effect, as fish are at a higher trophic level than molluscs and macroalgae.

The trophic transfer factor, which was not considered in this work, in many cases allows one to assess the potential for metal accumulation in the food chain, showing both biomagnification (TTF > 1) and biodilution (TTF < 1) [14]. Regarding the high metal concentrations in the fish from study estuary, these trophic transfers can be studied further, which will probably show a significant biomagnification for this estuary. There are many known examples of metal concentration relationships between predator and prey [14,55,56]. For example, the content of metals in the liver of a four-horned goby feeding on the crustacean *Saduria entomon* correlated with the content of metals in its tissues [56].

The physiological state of molluscs kept in cages was assessed by two functional characteristics, such as cardiotolerance and metabolic activity (oxygen consumption rate). After exposure to cages, they gave generally similar estimates. However, the high sensitivity of zebra mussels to natural environmental factors makes them a less convenient test object for monitoring with the cage method than unionid molluscs. In particular, the high mortality of dreissenid molluscs in cages at some sites can testify to unforeseen conditions for this species (high TP and oil compounds at sites 6 and 10). Furthermore, the HR of bivalve molluscs in U. pictorum responds to a variety of environmental factors, such as temperature, salinity, trophic factors, and pollution [57–60]. Close relationships were found between PAHs and HRR of unionid molluscs, i.e., at higher levels of 16 PAHs in water and increased HRR of molluscs after cage exposure in the area of oil port (St. 3).

Metabolic intensity, determined by the level of oxygen consumption, is also affected by natural factors, primarily temperature, but this indicator can be indicative in determining the effects of potential environmental toxicity [61,62]. Aerobic metabolism is sensitive to elevated metal concentrations [61]. It has been previously shown that the level of oxygen consumption in aquatic animals often decreases during acute exposure to high concentrations of toxic metals [62–64]. In our study, oxygen consumption rate in dreissenid molluscs exposed near hot-spot sites (port, dam, and eutrophied bay) was recorded as increased compared to reference group. A high level of oxygen consumption may be associated with the active excretion of oil products and other contaminants from the organism of molluscs and needs for oxidization the pollutant [65].

The presence of contaminants in the environment (mainly oil compounds) led to increased energy demands of animals (*Dreissena*) and reduced cardiac tolerance to stress (*Unio*); both can indicate about unfavourable conditions for molluscs and functioning of their organism near the border of active metabolism. The impact of pollution on molluscs can aggravate the condition of water-breathing animals, increasing their demands in oxygen, and when it is insufficient, molluscs reduce activity, close their shells, and switch to anaerobic metabolism [65,66]. Decreased heart rate is evidence for decreased activity of molluscs. A sharp decrease in heart rate as well as wide fluctuations in cardiac activity of bivalve molluscs were observed in experimental exposure under elevated oil concentrations [66].

Benthic detritivores (amphipod, crustacean, *G. fasciatus*, deposit-feeder, bivalve mollusc, *Unio pictorum*) were used in this paper as test species. The choice of molluscs and amphipods as suitable test species was determined by the criteria for the indicator organisms, primarily availability in the natural environment (abundant species), belonging to low links of the trophic web (first-level consumers), easy use in the laboratory, sensitivity, cost-efficiency, and preliminary development of the endpoints used. The classical method of biotesting based on the survival of crustaceans remains a very effective method [28]. Benthic amphipods have great potential in sediment toxicity testing because they are closely associated with sediment, either through their burrowing activity or ingestion of sediment particles [28]. Bivalves are sessile filter feeders that accumulate pollutants in their tissues and are widely used in pollution biomonitoring [67,68].

Unionid molluscs, in general, were most sensitive to oil pollution, while amphipods were used to complex contamination including heavy metal (cadmium and lead), organic contaminants (including PAHs), and eutrophication signs (high phosphorus and low oxygen). Such features may be associated with the vital activity and feeding habits of animals. Molluscs filter suspended solids from the water layer and the most superficial layer of bottom sediments, where, as a rule, oil products accumulate. Amphipods are able to burrow into bottom sediments and feed on sediment particles containing metals and others, thus accumulating them [28].

The results of the indication of natural biotopes polluted by metals and oil products, according to the applied indices in molluscs and amphipods, were comparable, and the use of multiple statistics methods made it possible, based on measurements, to identify several hot-spot areas with an increased risk of pollutant accumulation in the eastern Baltic Sea. In this case, the environmental assessment at selected contrast points in the Neva estuary was a good example to illustrate the methodological approach that can be applied in estuarine and coastal areas of various marine regions.

## 5. Conclusions

Bivalve molluscs, unionids, used in caging exposure show themselves as ideal bio-indicators in this study, as in previous research [13,69], mainly because of their capability to accumulate hazardous substances (metals, PAHs) and their high sensitivity to unfavourable factors. They are often used as test species in exposure experiments to various stressors. The present study shows that estimates based on the combined measurement of a set of parameters reflecting various functions of marine organisms provide good diagnostic power for the risk assessment of pollutants in the environment. Applied approaches such as the parallel use of bioaccumulation indices and biological biomarkers testifying the energy metabolism of the organism allow us to reveal strong relationships between the endpoints used with metal and oil pollution, which are very useful tools to detect these pollutants using the physiological state of biota. In general, the use of the triadic method (bioassay, bioaccumulation, and biomarkers) is a very promising approach and may become the standard in the future for monitoring of pollution (by metal and oil products) in marine environments. Using the caging method with transplanted molluscs is especially important in an estuary ecosystem with naturally low biodiversity.

**Supplementary Materials:** The following supporting information can be downloaded at https://www.mdpi.com/article/10.3390/jmse11091756/s1: Figure S1: Method of active monitoring was applied for environmental assessment using biomarkers of transplanted animals. View of cage with bivalve molluscs before exposure and after exposure; Figure S2: 15 individuals of *Unio* and 50 individuals of *Dreissena* were put in each cage for one month exposition in the Neva estuary; Figure S3: Unionid molluscs prepared to connection with the photopletismograph to measure heart rate. Special molecular glue used to attach adapter saddle for fibre-optic wire; Figure S4: Measurement of heart rate in tested molluscs; one Equipment complex allows us to measure eight molluscs at one time; Figure S5: Metal accumulation index: (a) caged and field molluscs; (b) field macroalgae; Table S1: Mortality (%) of molluscs *Dreissena polymorpha* and *Unio pictorum* after cage exposure in the Neva estuary; Table S2: Spearman rank order correlations; Table S3: Concentrations of PAHs (mkg/kg) in sediment at study sites.

**Author Contributions:** Conceptualization, N.B.; methodology, N.B., Y.G. and S.K.; software, N.B.; validation, A.M.; formal analysis, N.B.; investigation, N.B., A.M., A.S. and Y.G.; resources, A.M. and S.K.; data curation, N.B. and A.M.; writing—original draft preparation, N.B.; writing—review and editing, N.B., A.M., A.S. and Y.G.; visualization, N.B. All authors have read and agreed to the published version of the manuscript.

**Funding:** This research was funded by Ministry of Science and Education of the Russian Federation, grants 122031100274-7 (N.B., Y.G., and A.M.) and 122041100085-8 (A.S. and S.K.) and the Russian-Estonian Cross-Border Cooperation Program, grant ER90 HAZLESS (all authors).

**Institutional Review Board Statement:** The animal study protocol was approved and by the Ethics Committee of the Zoological Institute, the Russian Academy of Sciences (protocol codes 0.07, 14 February 2020 and 005, 9 April 2020).

**Informed Consent Statement:** Not applicable.

**Data Availability Statement:** The data presented in this study are available in Supplementary Materials.

**Acknowledgments:** We acknowledge Ivan Kuprijanov for discussion of the first results of the study.

**Conflicts of Interest:** The authors declare no conflict of interest.

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
