# Peer review of "Environmental Assessment with Cage Exposure in the Neva Estuary, Baltic Sea: Metal Bioaccumulation and Physiologic Activity of Bivalve Molluscs"

_jmse, doi:10.3390/jmse11091756_

Round 1

Reviewer 1 Report

Comments to the Authors

The paper ‘Environmental assessment with cage exposure in the Neva estuary, Baltic Sea: metal bioaccumulation and physiologic activity of bivalve molluscs’ aims to propose a comparative assessment of the environmental status at the Neva Estuary based on three axes : using bioaccumulation, bioassays and physiological biomarkers (cardio-tolerance and metabolic activity). The results were valuable and supported the good applicability of this ternary approach. Such an approach is significantly associated to a novelty in terms of the recognition of active and passive biomonitoring methods of the estuary. The manuscript deals thus with an important topic relevant to the environment and humans. We need such an integrative approach to assess the health status of any explored aquatic area. The paper is well-written and the language is of good quality. In my opinion, the writing manner is very effective in the uptake of prominent and key points of each research work, herein, the presentation way of the manuscript is very well and simple. The background provides a sufficient literature review, the methodology is sound, results are explanatory and well-discussed. Overall, a good read.

Abstract, introduction, and discussion:

These sections are well-written and focused on the main questions the authors intend to explore. In these sections, the authors explained too why they used the applied approach and what are their findings which could, in my opinion, improve the quality of future studies in biomonitoring of the aquatic ecosystems. The statistical side must be introduced/modified/rectified to support appropriately the trends observed.

To sum up, I found the paper very original and interesting, so I support its publication in the JMSE after a revision focucing especially on the statistical processing that I think might increase a the quality of the manuscript. All my detailed comments and suggestions are reported below:

Specific comments :

-line 139: the unit of salinity is PSU.

-line 306: Mann–Whitney U test : is it the right test to be used ? may be Dunn test is more correct since you are comparing first many numerical series by Kruskall Wallis ANOVA.

-line 311 : since you are transforming the data, why you used non parametric analyses : K-W and spearman ?? May be it’s better to transform all and use parametric tests : ANOVA and Pearson

-lines 305-315 : what softwares you used to perform the data processing ? should be given.

- Figure 3 (lines 387-390) : please indicate that the data were transformed.

-Line 418 : mkg ??

-line 538 : please add a reference. Please add more references .. sometimes you are discussing without citing references.

- line 577 : must be U. pictorum.

-References : use the same style for all references and use of MPDI.

Enough good

Author Response

Authors cordially thanks the Reviewer for valuable comments, recommendations and corrections.

Below we provided replies to Reviewer's detailed comments.

Reviewer:

-line 139: the unit of salinity is PSU.

Response: That is true that here salinity is sum of salts because of estuarine water was dissolved by freshwater of river and varied between fresh and brackish. The addition of PSU as a unit after the value is "formally incorrect and strongly discouraged acording to Pawlowicz, R. (2013). "Key Physical Variables in the Ocean: Temperature, Salinity, and Density"Nature Education Knowledge4 (4): 13. To clarify we changed salinity by " salt content" and left unit g/L here (lowest level is characteristic for real freshwater).

Reviewer:-line 306: Mann–Whitney U test : is it the right test to be used ? may be Dunn test is more correct since you are comparing first many numerical series by Kruskall Wallis ANOVA.

Response: Mann–Whitney U test is used as post hoc comparison  for Kruskall - Wallis test in PAST Statistics (that used for it). The Mann-Whitney-Wilcoxon test (also referred as Wilcoxon rank sum test or Mann-Whitney U test), used to compare two independent samples. This test is the non-parametric version of the Student’s t-test for independent samples. It is important also because we made comparison variables firstly with control(reference) point. 

Reviewer:-line 311 : since you are transforming the data, why you used non parametric analyses : K-W and spearman ?? May be it’s better to transform all and use parametric tests : ANOVA and Pearson

Response: Number of observation (n <30). Number of observations was below required limit that allow us use parametric statistics. Data standartization was used for multicomparisons as PCA analisys.

Reviewer:-lines 305-315 : what softwares you used to perform the data processing ? should be given.

Response: we added missing information. Past 4 software and Statistica 10.0 software were used.

Reviewer:- Figure 3 (lines 387-390) : please indicate that the data were transformed.

Response: data was transformed before PCA. Lines 313-317 explain these manupulations. The data were log-transformed, centred, and normalised to avoid misclassification due to differences in data dimensions. Zero mean and unit variance (z-scores) were normalised by subtracting their mean from each value and then dividing by their standard deviation. These z-scores were further used in correlation matrices in multivariate analysis (including PCA). we added "including PCA" for clarity.

Reviewer:-Line 418 : mkg ??

Response: Yes, we added missing unit. μg/L

Reviewer:-line 538 : please add a reference. Please add more references .. sometimes you are discussing without citing references.

Response: Multifactor influence on Cd bioaccumulation were also found earlier [16,49]. we also added more references to discussion part.

Reviewer:- line 577 : must be U. pictorum.

Response: It was corrected

Reviewer:-References : use the same style for all references and use of MPDI.

Response: References were revised following to MDPI Style.

Reviewer 2 Report

The study investigates the effects of metal bioaccumulation and PAH on the biota (bivalves, macroalgae, and amphipods) in the Neva Estuary, Baltic Sea. The article must be improved on objectivity since the hypothesis tested is unclear. It needs to be clarified if the novelty methodological approach suggested by this study effectively identifies the correlation between pollution and its effects on the biota. Before this manuscript can be considered for publication, the authors need to address several aspects:

The introduction could be improved on the logical flow. The transition between the introductory statements and the discussion of estuarine ecosystems could be smoother. Consider rephrasing to make the connection more straightforward. For example, after discussing the rise of anthropogenic impacts, you could directly introduce estuaries as ecosystems vulnerable to these impacts, with a brief explanation of their transitional nature. Consider separating information about the types of heavy metals, their sources, accumulation, and potential effects into distinct paragraphs. The introduction also fails to establish a more straightforward link between exposure to pollutants and using mollusks as indicators. Provide context for "Biosensor Systems" - When introducing the concept of biosensor systems, explain these systems and how they relate to the study's objectives.

The methods section should clarify the choice of the sampling points and the July-August period for observations and explain why these months were chosen for the fieldwork. Additionally, provide reasoning for using the 1-month exposure period for caged mollusks. When describing the 10-day survival tests with crustaceans, provide more context about Gmelinoides fasciatus and its ecological relevance in the study area. Provide a concise rationale for measuring mollusk physiological states (cardioactivity and metabolic rate). Explain why these parameters were chosen and how they relate to the research question.

The discussion section must be improved in several aspects:

o   The assertion that organic pollutant accumulation is slower than that of metals could benefit from more context. Please explain why this difference in accumulation rates might occur and how it aligns with existing knowledge in the field.

o   While it's mentioned that macroalgae are good indicators, provide more insight into how their transition from sediment to food webs impacts the overall ecosystem. Discuss their role in the larger context of the estuarine environment and why they are relevant as indicators.

o   When discussing the regional standards used in the study, briefly address the potential limitations or uncertainties associated with using these standards and how they might affect the interpretation of the results.

o   While you describe the distribution of metals in the coastal zone, consider providing a more straightforward link between the distribution patterns and their potential environmental consequences. How do the observed concentrations relate to potential ecological risks or impacts on organisms? Also, discuss how they might affect the health of mollusks and the organisms that feed on them.

o   Explain why the variability in metal accumulation occurs. Are specific environmental or biological factors contributing to mollusks' observed range of metal concentrations?

o   Link organisms' elevated metabolic rates and heart rates to the specific environmental conditions or pollutants observed in the study. Discuss the potential implications of these responses for the health and survival of the studied species.

o   Finally, and more importantly, briefly address the broader contributions of this study to the field. How do the findings advance our understanding of pollutant impacts and environmental health in estuarine ecosystems?

Line 38 – Remove “developing.”

Line 44 – Change “the assessment of” to “assessing.”

Line 52 – Change “taking into account” to “considering.”

Standardize the use of the term mollusc or mollusk.

There are several grammar issues in the text that must be addressed.

Author Response

Reviewer:

  1. The study investigates the effects of metal bioaccumulation and PAH on the biota (bivalves, macroalgae, and amphipods) in the Neva Estuary, Baltic Sea. The article must be improved on objectivity since the hypothesis tested is unclear. It needs to be clarified if the novelty methodological approach suggested by this study effectively identifies the correlation between pollution and its effects on the biota. Before this manuscript can be considered for publication, the authors need to address several aspects:

 Respond: It was necessary to find out whether the new methodological approach (ie the use of a triadic method: biotesting with crustaceans, bioaccumulation indices and physiological state of transplanted mollusks) will effectively determine the correlation between pollution and the consequences of its impact on biota.

Reviewer: 2.The introduction could be improved on the logical flow. The transition between the introductory statements and the discussion of estuarine ecosystems could be smoother. Consider rephrasing to make the connection more straightforward. For example, after discussing the rise of anthropogenic impacts, you could directly introduce estuaries as ecosystems vulnerable to these impacts, with a brief explanation of their transitional nature. Consider separating information about the types of heavy metals, their sources, accumulation, and potential effects into distinct paragraphs. The introduction also fails to establish a more straightforward link between exposure to pollutants and using mollusks as indicators. Provide context for "Biosensor Systems" - When introducing the concept of biosensor systems, explain these systems and how they relate to the study's objectives.

Respond: Intro was improved, as recommended by reviewer.

Reviewer: The methods section should clarify the choice of the sampling points and the July-August period for observations and explain why these months were chosen for the fieldwork. Additionally, provide reasoning for using the 1-month exposure period for caged mollusks. When describing the 10-day survival tests with crustaceans, provide more context about Gmelinoides fasciatus and its ecological relevance in the study area. Provide a concise rationale for measuring mollusk physiological states (cardioactivity and metabolic rate). Explain why these parameters were chosen and how they relate to the research question.

Respond: We added explanations. The duration of experiment was chosen basing on previous results [25] to obtain significant level of metal bioaccumulation. The month of experiment was selected as temperature- and food-comfort period for both species of molluscs since bioaccumulation of pollutants depends on temperature and trophic conditions [21]. Rationale for measuring mollusk physiological states (cardioactivity and metabolic rate) are explained in Introduction.

Reviewer: The discussion section must be improved in several aspects:

Respond: It was improved. All changes were shown in colored text.

o   The assertion that organic pollutant accumulation is slower than that of metals could benefit from more context. Please explain why this difference in accumulation rates might occur and how it aligns with existing knowledge in the field.

o   While it's mentioned that macroalgae are good indicators, provide more insight into how their transition from sediment to food webs impacts the overall ecosystem. Discuss their role in the larger context of the estuarine environment and why they are relevant as indicators.

o   When discussing the regional standards used in the study, briefly address the potential limitations or uncertainties associated with using these standards and how they might affect the interpretation of the results.

o   While you describe the distribution of metals in the coastal zone, consider providing a more straightforward link between the distribution patterns and their potential environmental consequences. How do the observed concentrations relate to potential ecological risks or impacts on organisms? Also, discuss how they might affect the health of mollusks and the organisms that feed on them.

o   Explain why the variability in metal accumulation occurs. Are specific environmental or biological factors contributing to mollusks' observed range of metal concentrations?

o   Link organisms' elevated metabolic rates and heart rates to the specific environmental conditions or pollutants observed in the study. Discuss the potential implications of these responses for the health and survival of the studied species.

o   Finally, and more importantly, briefly address the broader contributions of this study to the field. How do the findings advance our understanding of pollutant impacts and environmental health in estuarine ecosystems?

 Quality of English Language was improved

We cordially thank the reviver for valuable comments.

Reviewer 3 Report

Well written, easy to read, objectives very clear.

The aim of the work was to carry out a comparative health assessment in the Neva Estuary (Gulf of Finland) using physiological biomarkers, biotesting
and bioaccumulation. in different sentinel organisms. They designed biomonitoring of the area combining both active and passive methods.

Minor mistakes and comments:

.- Line 91: environ-mental assessment

.- Line 318: salinity and other variables data are not presented in Table 1 but in Table 2. Table 1 is for quality criteria.

.- Which are the PAHS measured? In tbale 2 is said that are 16 EPA, but, please, specify in material and methods section.

.- I guess that in table 3 all the concentrations of metals over the MAC range are written in bold. This should be stated in the legend of the table.

.- Figure 2. Recoomendation: Use the same scale to compare both data visually and quickly.

.- Line 365. Include table 4 or Figure 3  after BCF. Why to use both, the table and the graphs? One way of illustrating the results might be enough.

.- Style: Reccomendation to use the same style of graphs (Figure 3 vs Figure 4 vs. Figure 5 vs figure 6): colour, thickness of lines.....

:- Lack of papers regarding cell and tissue level biomarkers in molluscs from the Baltic se (paper by Benito et al. 20198 (STOTEN). Maybe, same baseline values (biomarkers, accumulationof pollutants) for mussels stated in that paper can be useful for comparison purposes.

.- The conclusion (lines 597-603) should be stated as conclusion, at the beginning of the paragraph.

Author Response

Thank you for valuable comments and proposed corrections.

We changes Lines 91. 

Line 318 was corrected. Table 1 was removed, quality criteria put directly in the text, so now table1 is about salinity and other variables

The PAHs measured was listed in methods and Supplementary material. 

Legend of Table 3 was corrected.  All the concentrations of metals over the MAC range are written in bold.

Table 4 was left and Figure 3 was  moved to Supplementary material.

Figure 4 vs. Figure 5 vs figure 6 are now in the same style, figure 3 was removed.

Paper regarding cell and tissue level biomarkers in molluscs from the Baltic sea was added to Reference and discussion. 

The conclusion was separated as the paragraph.

Round 2

Reviewer 2 Report

The authors addressed all issues in the manuscript. Nice job!